# Addressing Genetic Tumor Heterogeneity, Post-Therapy Metastatic Spread, Cancer Repopulation, and Development of Acquired Tumor Cell Resistance

**DOI:** 10.3390/cancers16010180

**Published:** 2023-12-29

**Authors:** Dennis Christoph Harrer, Florian Lüke, Tobias Pukrop, Lina Ghibelli, Albrecht Reichle, Daniel Heudobler

**Affiliations:** 1Department of Internal Medicine III, Hematology and Oncology, University Hospital Regensburg, Franz-Josef-Strauß-Allee 11, 93053 Regensburg, Germany; dennis.harrer@ukr.de (D.C.H.); florian.lueke@ukr.de (F.L.); tobias.pukrop@ukr.de (T.P.); daniel.heudobler@ukr.de (D.H.); 2Division of Personalized Tumor Therapy, Fraunhofer Institute for Toxicology and Experimental Medicine, 30625 Regensburg, Germany; 3Bavarian Cancer Research Center (BZKF), University Hospital Regensburg, 93053 Regensburg, Germany; 4Department of Biology, University of Rome “Tor Vergata”, 00133 Rome, Italy; ghibelli@uniroma2.it

**Keywords:** tumor tissue editing, anakoinosis, M-CRAC, drug resistance, tumor heterogeneity, tumor cell repopulation, metastases, pioglitazone, drug repurposing, metronomic chemotherapy, transcriptional modulation

## Abstract

**Simple Summary:**

The present review summarizes and interprets uniform therapy schemes for rescuing relapsed or refractory (r/r) neoplasias of quite different histologic origins. Exploiting tumor tissues’ plasticity by reprogramming hallmarks of cancer into biologic hallmarks controlling tumor regrowth with metronomic chemotherapy and simultaneous targeting of nuclear and/or cytokine receptors plus/minus targeted therapies, termed tumor tissue editing, may induce cCR or long-term tumor control, as indicated by data from clinical trials designed for the treatment of r/r neoplasias of quite different histologic origins. Tumor tissue editing may overcome unique post-therapy disease traits that arise following treatment of r/r tumor disease with standard therapy regimens using maximum tolerable doses, i.e., metastatic spread, cancer repopulation, and acquired tumor cell resistance (M-CRAC), by attenuating, or resolving M-CRAC. The introduction of M-CRAC control in future therapeutic considerations may help to overcome the multifold translational challenges of precision medicine in the large group of r/r neoplasias without driver mutations.

**Abstract:**

The concept of post-therapy metastatic spread, cancer repopulation and acquired tumor cell resistance (M-CRAC) rationalizes tumor progression because of tumor cell heterogeneity arising from post-therapy genetic damage and subsequent tissue repair mechanisms. Therapeutic strategies designed to specifically address M-CRAC involve tissue editing approaches, such as low-dose metronomic chemotherapy and the use of transcriptional modulators with or without targeted therapies. Notably, tumor tissue editing holds the potential to treat patients, who are refractory to or relapsing (r/r) after conventional chemotherapy, which is usually based on administering a maximum tolerable dose of a cytostatic drugs. Clinical trials enrolling patients with r/r malignancies, e.g., non-small cell lung cancer, Hodgkin’s lymphoma, Langerhans cell histiocytosis and acute myelocytic leukemia, indicate that tissue editing approaches could yield tangible clinical benefit. In contrast to conventional chemotherapy or state-of-the-art precision medicine, tissue editing employs a multi-pronged approach targeting important drivers of M-CRAC across various tumor entities, thereby, simultaneously engaging tumor cell differentiation, immunomodulation, and inflammation control. In this review, we highlight the M-CRAC concept as a major factor in resistance to conventional cancer therapies and discusses tissue editing as a potential treatment.

## 1. Introduction

The treatment of metastatic tumors poses a crucial challenge in anti-cancer therapy. Although the therapeutic armamentarium has steadily grown and overall survival (OS) rates have significantly improved, continuous complete remissions (cCRs) are still scarce in patients with refractory or relapsing (r/r) metastatic tumor disease [1,2,3,4,5,6,7]. 

Conventional tumor therapy based on the application of a maximum tolerable dose to achieve maximum apoptosis induction in cancer cells is still a vital pillar of anti-cancer therapy [1,6,8]. However, disease relapse after initially successful chemotherapy is frequently observed, and relapsing cancer cells often display elevated resistance to traditional chemotherapy schedules [1]. 

Seeking to explain tumor progression after conventional tumor therapy, the concept of post-therapy metastatic spread, cancer repopulation, and acquired tumor cell resistance (M-CRAC) is primarily predicated on tumor cell heterogeneity generated as a result of post-chemotherapy genetic damage and subsequent tissue repair mechanisms [9,10]. Due to M-CRAC processes, tumor cells and the associated tumor microenvironment give rise to significant resistance and pave the way for r/r metastatic disease [8,11]. 

Contrary to conventional chemotherapy, immunotherapy or targeted therapies with small molecules, tumor tissue editing approaches therapeutically design the tumor phenotype by simultaneously engaging tumor cell differentiation, immunosurveillance, inflammation, and tumor metabolism via redirection of cancer hallmarks. These factors are important drivers of M-CRAC in a multitude of histologically different relapsed or refractory tumor entities treated with tumor tissue editing approaches [12,13,14,15,16,17,18,19,20,21,22,23,24,25,26,27,28,29,30,31].

Tissue editing approaches encompass low-dose metronomic chemotherapy, the use of transcriptional modulators, and targeted therapies. Clinically, treatment regimens based on tissue editing can achieve continuous complete remissions in patients with a variety of different r/r malignancies, such as Hodgkin’s lymphoma, Langerhans cell histiocytosis, renal clear cell carcinoma, cholangiocarcinoma, and angiosarcoma. This definitively shows that M-CRAC induced by preceding systemic therapies may be overcome, even by induction of cCR in single patients with histologically different r/r tumor disease [13,16,24,25,26].

In the present review, we introduce the M-CRAC concept as a key factor contributing to clinical resistance to standard systemic tumor therapies. In addition, we highlight tumor tissue editing strategies as specific interventions to counteract M-CRAC and maintain long-lasting disease control. 

## 2. M-CRAC as a Target for Tumor Tissue Editing in Refractory or Relapsed Neoplasias

The novel treatment paradigm ‘tumor tissue editing’ adopts the use of tissue editing technologies for correcting genetic or epigenetic abnormalities in tumor tissues [32,33,34,35,36]. Tumor tissue editing methodologies are now focused on achieving phenotypic, therapeutically relevant editing of tumors [37].

Tumor tissue editing is defined as the therapy-guided targeted evolution of tumor tissues to establish biologic hallmarks that facilitate tumor control or initiate complete remission in relapsed or refractory tumor disease [38]. Appropriate editing schedules combine bioactive therapeutic principles to readjust aberrant tissue homeostasis at primary and metastatic tumor sites by using differential regulatory active therapeutic techniques [39,40]. Bioactive drugs must not demonstrate any monoactivity in the respective tumor disease [40]. The combined regulatory activity profile concertedly redirecting hallmarks of cancer is of pivotal therapeutic interest (Table 1).

The operational procedure facilitating tumor tissue editing, i.e., anakoinosis, promotes therapy-initiated reprogramming and cell recruitment in tumor tissues, which finally contributes to the tumors’ plasticity in therapeutic intention [39,40]. Tumor-associated supervising communication lines that cumulatively constitute tumor-type specific communication protocols among different tumor cell compartments are supposed to be respective therapeutic targets of editing techniques [40,48]. 

The editing procedure relies on tumor-specific communication lines to establish phenotypic plasticity. The extent and quality of tumor systems’ plasticity reflect evolutionary system states and the developmental and medical histories of tumor diseases [12,44,49,50]. Tumor tissue plasticity is constituted by timely and spatially developing tumor cell autonomous and non-autonomous processes and cannot be described by genetic/molecular genetic aberrations alone, particularly if driver mutations are absent, as in most tumors [51,52,53]. 

Tumor tissue editing approaches turned out to efficaciously control or resolve M-CRAC by redirecting cancer associated hallmarks into biologic hallmarks, attenuating tumor growth, and inducing alternative patterns of tumor cell death in r/r tumor disease. As shown, clinical trials using editing approaches may induce long-term tumor control, objective response or even cCR [38] (Figure 1, Table 1).

The novel treatment strategy is highly integrative and provides, beyond its biomodulatory activity profile, opportunities for drug repurposing, e.g., by providing edited non-oncogene addiction. Moreover, editing techniques may establish transdifferentiation and differentiation of tumor cells for M-CRAC control and tumor cell death induction [12,37,49]. 

Important aspects of tissue editing approaches include the possibility of exploiting the synergistic potency and efficacy of biomodulatorily active drug combinations [54]. The combinatorial, metronomic administration of drugs targeting the network infrastructure of tumor tissues proved to be efficacious in various, histologically quite different tumors. Interestingly, edited tumor tissues provide novel activity profiles for otherwise less efficacious approved targeted therapies. As clinically shown, the use of repurposed targeted therapies, e.g., mTOR inhibitors, may contribute to long-term tumor control or cCR in r/r tumor disease, e.g., Hodgkin’s lymphoma or uveal melanoma [22,37].

Tumor tissue editing approaches for rescuing r/r neoplasias via M-CRAC control and tumor cell death induction may include different metronomically scheduled cytotoxic drugs (trofosfamide, treosulfan, capecitabine). Low-dose metronomic chemotherapy serves as a prerequisite for the activity of otherwise inefficacious transcriptional modulators [40]. Tissue editing approaches use transcriptional modulators, such as nuclear receptor agonists or cytokines, as indicated in Table 1. Pioglitazone itself is a dual-receptor agonist for the peroxisome-proliferator-activated receptor PPARα and γ (PPARα/γ). Therefore, all the mentioned tumor tissue editing trials used dual or triple transcriptional regulation in addition to low-dose metronomic chemotherapy (Table 1) [55].

With tissue editing techniques, quite different neoplasias in the r/r stage could be either stabilized for the long term or controlled by objective response, even by cCR [13,16,19,22,25,26]. Further, in castration-resistant prostate cancer (CRPC) or r/r multiple myeloma (MM), disease control was stable even after discontinuation of the tumor editing approach. In CRPC, hormone sensitivity was reestablished in single cases [19]. Stable M-CRAC control beyond discontinuation of tumor tissue editing in CRPC and r/r multiple myeloma reveals reprogramming of the tumor tissue, i.e., simultaneously involving the stroma and tumor cells [14,19].

Thus, tumor tissue editing techniques facilitate the achievement of quite different endpoints that finally contribute to M-CRAC control. Objective response is not a prerequisite for M-CRAC control [14]. 

## 3. Tissue Editing Methods Redirect Cancer-Related Hallmarks into Novel Biologic Hallmarks Facilitating Tumor Response 

How tissue editing approaches work at the tumor tissue communication level cannot yet be pinned down to single pathways [40]. 

Studies on non-PML-AML provide the insight that only the triple combination of azacitidine, ATRA, and pioglitazone may induce cytologically proven differentiation as a triple combination. These results, however, underline that the synergism of each drug with one another is a prerequisite for therapeutically redirecting leukemia plasticity [45,50].

Also, clinical data on r/r RCCC, showing cCR induction with the combination pioglitazone plus interferon-α on the background of ‘very’-low-dose capecitabine, are not well experimentally supported [25]. However, reprogramming the oncogenic stress response is the prerequisite for dual control of tumor-associated inflammation with pioglitazone and interferon-α. C-reactive protein (CRP) is a strong marker indicating tumor response [17,43]. In RCCC, tumor cells may directly produce CRP aside from liver-mediated secretion [56]. Different tissue editing approaches, as indicated in Table 1, result in diversified reprogramming results of tumor tissues. 

Also, the recently published survival benefit of patients with relapsed head and neck cancer receiving metronomic methotrexate, celecoxib, and erlotinib combined with low-dose nivolumab in comparison to those receiving the same metronomic schedule without nivolumab cannot be based on pathway analyses providing distinct mechanisms of action promoting OS benefit with quadruple therapy [57]. Enhancement of the immune response might be the main emphasis of the nivolumab arm. 

In general, the studies on tissue editing demonstrate the efficacy of tumor editing techniques in selectively redirecting cancer hallmarks for tumor control.

Within tissue editing approaches, pleiotropic tissue activity profiles of single biomodulatory drugs lead in concerted tissue communication guiding activity profiles that are focused and successfully reprogram the oncogenic stress response as a prerequisite for redirecting single cancer hallmarks into biologic hallmarks attenuating tumor growth [40]. This way, tumor tissue editing approaches therapeutically exploit tumor plasticity (Figure 1). The reprogramming process is termed anakoinosis in anticipation of future descriptions of communicative lines describing the molecular background of the respective pharmacologic interactions on the tissue level in more detail [40]. The activity profiles of metronomic chemotherapy, pioglitazone, and other transcriptional modulators in reviews did not predict the outcome provided by tissue editing protocols that have been successfully applied in many histologically highly different neoplasias [55,58,59].

## 4. Tissue Editing Approaches: Impact on the Genomic Evolution of Tumors

A remarkable clinical result is that tumor tissue editing approaches may induce CR or cCR in r/r neoplasias, even within small study populations (Table 1). 

Recent studies found molecular genetic markers or histologies associated with resistance to metronomic chemotherapy in head and neck cancer or triple-negative breast cancer [60,61]. These data might not be directly transferable to tissue editing approaches. Metronomic low-dose chemotherapy facilitates the use of the clinically decisive step, concerted transcriptional modulation. As shown in non-PML AML, even complex aberrant karyotypes may respond with CR to editing methods [45]. 

Hodgkin’s lymphoma in the relapsed or refractory stage demonstrates that an adequate tissue editing approach overcomes resistance mechanisms of quite different origin, namely those induced by radiotherapy, immunotherapy, and chemotherapy, as indicated by consecutive CR and cCR induction [13]. Thus, editing-specific genomic evolution of resistance plays no key role when using an adequate editing methodology.

Nevertheless, developing resistance may limit response to tissue editing approaches in r/r tumor disease. Up to now, there was no urgent need to perform genetic analyses at the time of tumor progression, because there are fewer concerns about the rapid evolution of resistance to tissue editing approaches. Currently, the main therapeutic focus is the implementation of the most suitable editing approach, at best in a personalized manner. Further studies must establish missing biomarkers for the personalized selection of editing techniques and corresponding follow-up parameters. The tissue editing methodology is at its beginning (Figure 2). 

## 5. M-CRAC as Driver of Disease Relapse and Chemoresistance: The Therapeutic View 

Definition of M-CRAC: M-CRAC-associated disease traits, i.e., metastatic spread, cancer repopulation, and acquired tumor cell resistance as well as genetic and/or molecular-genetic tumor cell heterogeneity, may be clinically separated as a unique post-therapeutic response pattern to systemic tumor therapies based on the clinical finding that M-CRAC may be successfully resolved or attenuated by the introduction of tumor tissue editing techniques designed for the treatment of r/r neoplasias of different histologic origin (Table 1) [12,13,14,15,16,17,18,19,20,21,22,23,24,25,26,27,28]. In particular, the differentially developing resistance patterns in tumors, originating either from resistant clones during initial tumor growth or from the multifold resistance patterns induced by preceding systemic tumor therapies, describe the therapeutic challenges for establishing M-CRAC control and indicate the necessity for novel therapeutic solutions. The multifaceted M-CRAC disease traits have been intensively studied, e.g., in r/rMM [62,63,64,65].

The M-CRAC concept summarizes disease traits promoting tumor progression or relapse following any kind of prior therapy, such as chemotherapy, immunotherapy, and targeted therapies with small molecules. M-CARC is often associated with mixed or organ-specific response patterns to systemic tumor therapy [9,10,66,67]. 

The availability of specific tumor-type-adapted tissue editing techniques for rescuing relapsed or refractory tumor disease of quite different histologic origins support the consistency of the M-CRAC concept (Table 1).

Biologic processes promoting M-CRAC: Intrinsic, i.e., systemic, and tumor-specific factors guide the biology of metastatic tumor disease. Tumor-cell-specific parameters are commonly considered targets for systemic tumor therapy to attain maximum apoptosis induction [68]. Additionally, extrinsic, primarily non-genetic factors, termed the ‘exposome’ of the host, impact tumor biology and long-term outcome following systemic tumor therapy and play a pivotal role in the development of M-CRAC [8,69,70]. All preceding chemotherapies, immunotherapies, targeted therapies with small molecules, and radiation therapies contribute to the ‘exposome’ and are directly involved in the development of M-CRAC. In aggregate, M-CRAC describes the oncogenic potential of prior tumor therapies promoting pathophysiological disease traits associated with tumor progression or relapse [8,9]. 

Potential mechanisms underlying M-CRAC: Damage response to systemic tumor therapy reshapes cancer tissue by a multitude of coordinated, timely, spatially, and dynamically developing cellular and tissue events, summarized in M-CRAC [9,71]. The enormous range of post-therapy tumor system states, arising on the basis of the tumor tissue’s plasticity, facilitates the maintenance of the tumor tissue’s integrity via wound healing mechanisms [72]. Multifold concurrent events linked to M-CRAC-related disease traits are closely interwoven and underline the oncogenic role of tumor cell death during post-therapy tissue repair [72,73,74,75,76,77]. 

Apoptotic tumor cells, hypoxia, and inflammation may promote compensatory tumor cell proliferation, for instance through the caspase-3-cytosolic phospholipase A(2) alpha (cPLA-2)-COX-2-PGE-2-STAT3 Phoenix rising pathway. [10,68,78,79]. Maximum-tolerated doses (MTD) of chemotherapy may promote M-CRAC via the epithelial–mesenchymal transition (EMT) orchestrated by hypoxia-inducible factors (HIF)-1α and (HIF)-2α [79,80,81,82,83,84,85].

For future research, experimental studies are vital to investigate how the post-therapy oncogenic potential of pulsed therapies, summarized in M-CRAC-related disease traits, might be accessible with differential tissue editing approaches [10,11,86,87,88,89,90]. 

The therapeutic accessibility of M-CRAC via tumor tissue editing techniques is surprising, as tumor progression following systemic tumor therapy is spatially and temporally heterogeneous. Tumor cells are epigenetically remodeled and may successively become genetically heterogeneous. Moreover, stromal cells may gain genetic aberrations, cancer stem cells become functionally altered, non-mutated tumor suppressors are down-regulated, epithelial–mesenchymal transition is promoted, and autophagy becomes dysregulated. Finally, context-dependent roles of TGFβ occur, tumor and stroma cell metabolism are disturbed, immunosurveillance is decreased, pro-inflammatory processes in hypoxia are highly activated, and angiogenesis is altered [91,92,93,94,95,96,97,98,99,100,101,102,103,104,105,106,107,108].

As a result of the multi-leveled repair processes, the phenotypic outcome of M-CRAC seems to be poorly predictable. The cellular tumor tissue compartments exploit the whole repertoire of cellular plasticity to guarantee tissue integrity at the expense of M-CRAC development [109,110,111,112,113,114]. The evolution within cancer tissues to establish novel tissue homeostasis under hypoxic, pro-inflammatory, and immunosuppressive conditions and the presence of damaged cells involves all cellular compartments [3,109,115,116,117]. In the end, even secondary malignancies are possible as a final ‘reconstitution’ process.

The induction of cCR in r/r neoplasias indicates resolution of M-CRAC. Differentiation induction may completely bypass M-CRAC, as shown by hematologic or molecular-genetic complete remission in relapsed or refractory acute myelocytic leukemia (AML), even in those with adverse risk according to European LeukemiaNet (ELN) [12,45,46,47].

This means the clinically circumscriptive phenotype defined as M-CRAC is differentially accessible with tumor tissue editing techniques using biomodulatory techniques, i.e., anakoinosis, for successfully targeting the evolutionary preserved plasticity of tumor tissues in therapeutic intention (Figure 1). Different tumor histologies share targets available for M-CRAC control, even if in quite different contexts, as exemplified for pioglitazone (Table 1). 

An important mechanism of action for meeting the multifold therapeutic challenges provoked by M-CRAC might be the therapeutic alteration and redirection of the tumor-specific, survival-related inevitable stress response patterns via metronomic low-dose chemotherapy and, additionally, via targeted transcriptional modulation. The feasibility of establishing edited non-oncogene addiction to mTOR in r/r HL, followed by the achievement of cCR, supports the explanation that Hodgkin tissue cannot manage any more stress responses, irrespective of the kind of pre-treatment [13].

The available data on tissue editing trials cannot yet specify whether differential editing approaches are necessary for M-CRAC control to meet specific tissue alterations given by preceding systemic tumor therapies. However, different prior systemic targeted therapies, chemotherapies, and immunotherapies within one histologic tumor type were rescued with unique tissue editing techniques, as exemplarily shown in r/r acute myelocytic leukemia (AML) and r/r Hodgkin lymphoma (HL) [12,13,45,46,47].

Explaining M-CRAC by tumor tissue editing techniques: Tissue editing techniques therapeutically target the network infrastructure of tumor tissues and the tumor’s phenotypic plasticity, which contrasts with frequently used approaches separately targeting tumor and/or stroma cells (Figure 1). Therapeutical reprogramming of the tumor infrastructure promotes new ways of tumor cell death. Moreover, differentiation induction and post-therapy wound healing processes may be modulated to resolve or bypass M-CRAC [25,50,118,119,120]. The induction of cCR in r/r metastatic neoplasias shows that tumor cell heterogeneity in metastatic r/r disease may be overcome by functionally editing the whole repertoire of heterogeneous tumor phenotypes and genetically heterogeneous tumor cells [13,16,24,25,26]. Tissue editing techniques demonstrate, even in small study populations and in histologically quite heterogeneous neoplasias successful control of r/r metastatic disease (Table 1) [12,13,14,15,16,17,18,19,20,21,22,23,24,25,26,27,28].

Single pathways are not attributable as cause of complex post-therapy disease traits mediated by M-CRAC. Tissue editing techniques may address M-CRAC, which emerged as regulatory and differentially accessible phenomenon. M-CRAC control aims at reprogramming communication networks that are supported by tumor and stroma cells. 

## 6. Specific Therapeutic Access to M-CRAC with Tumor Tissue Editing Approaches

Therapeutic challenges in treating M-CRAC with chemotherapy, immunotherapy, or targeted therapies with small molecules: Enhancing apoptosis induction by the application of maximum tolerable doses is the main aim of most systemic tumor therapies. Irrespective of the drug combination, the drug-related ‘exposome’ leaves major therapeutic challenges, as reflected by a high frequency of relapsed or refractory diseases. Systemic tumor therapy rarely achieves long-term tumor control or cCR in most r/r metastatic neoplasias despite the steadily increasing and diversifying therapeutic repertoire. Reasons are tissue repair processes following apoptosis, hypoxia, cell damage, immunosuppression, metabolic alterations and inflammation [1,2,4,5,6,7,8,121].

M-CRAC is a major obstacle counteracting the initially induced tumor response and therefore affects the overall outcome. The approved pulsed chemotherapy regimens, relying on the application of maximum tolerable doses, primarily aim at tumor cell destruction, resulting in the stimulation of tissue repair and potential M-CRAC development [1]. 

Therefore, induction therapies designed to induce maximum rates of tumor cell death should be complemented with therapies targeting and guiding the tumor tissues’ post-therapy functional phenotypes to simultaneously address M-CRAC [83]. 

Differential techniques of tumor tissue editing uncover M-CRAC as a unique but differentially accessible pathophysiological phenomenon. Three therapeutic elements systematically addressed M-CRA: Low-dose metronomic chemotherapy, epigenetic modifiers (azacitidine), and transcriptional modulation via nuclear receptors (PPARα/γ glucocorticoid receptor, RXR, or cytokine receptors). Moreover, the combination with classic targeted therapies could be beneficial, as exemplarily shown for mTOR inhibitors by the edited non-oncogene addiction in r/r HL for mTOR (Figure 1, Table 1) [13,45]. 

Resolution of M-CRAC has been achieved as indicated by cCR in r/r cholangiocellular carcinoma, renal clear cell carcinoma, angiosarcoma, multisystem Langerhans cell histiocytosis, Hodgkin’s lymphoma, or by CR in non-promyelocytic acute myelocytic leukemia (non-PML-AML) [12,13,16,24,25,45,122]. Tissue editing induced long-term tumor control in all studied r/r neoplasias, in high grade gliomas stable disease as best response, and in all the other neoplasias at least partial remissions (Table 1).

Tumor plasticity is the starting point and final target for tumor tissue editing methods [9,37]. Differential therapeutic emphasis on distinctively developed tumor-associated hallmarks may help to select specific tumor tissue editing techniques (Figure 1).

All stroma components, i.e., mesenchymal cells, cancer-associated fibroblasts (CAFs), adipocytes, endothelial cells and hematologic cells, such as myeloid-derived suppressive cells, tumor-associated macrophages (TAMs) and T-cells, are involved in M-CRAC control, but to different extents depending on the focus of the chosen biomodulatory drug combination [55,59]. Enhancement of immunosurveillance seems to frequently play a crucial role in successful editing techniques, besides strong inflammation control, differentiation induction and metabolic reprogramming (Table 1, Figure 1) [23,42,57].

Metronomic low-dose chemotherapy as a prerequisite for transcriptional editing During long-term administration of metronomic low-dose chemotherapy in schedules for tumor tissue editing, scheduled dose reductions of cytotoxic drugs were allowed, even up to a quarter of the initially recommended starting dose [17]. Thus, the initially intended monoactivity of metronomic ‘very’ low-dose chemotherapy to induce at least PR or even cCR is out of range. 

The study on r/r renal clear cell carcinoma (RCCC) impressively shows the effect of ‘very’ low doses of capecitabine. The addition of low-dose interferon-α to pioglitazone and cyclooxygenase-2 (COX-2) inhibition may induce cCR on the background of very low-doses of capecitabine, namely, 1 g absolute BID × 2 weeks, followed by a 1-week rest period, and scheduled dose reductions of capecitabine to 0.5 g or 0.25 g absolute BID × 2 weeks, followed by a 1-week rest period. In contrast to the approved standard dose of capecitabine, 1250 mg/m^2^ BID within the same schedule, the starting dose of capecitabine in the phase II editing trials was 40% in comparison to the standard dose capecitabine, and 20% or 10% if scheduled dose reduction was performed in phase II editing trials for RCCC [17,43]. 

Metronomic, very low-dose chemotherapy promotes a continuous pattern of stress responses [55,123,124,125]. The present clinical data reveal that metronomic chemotherapy at ‘very’ low-doses limits tumor tissue plasticity by stress response, probably decreasing phenotypic heterogeneity of tumor cell niches as tissue stress generally induces a tighter phenotype [123,126,127,128,129,130,131]. Thus, metronomic chemotherapy might induce phenotypic integration of inflammation control or differentiation by editing techniques and, consecutively, may serve as an enhancer of pro-anakoinotic effects mediated by added transcriptional modulators [55,123]. Exposure of tumor tissues to stress in addition to the tumor intrinsic management of oncogenic stress for preserving tumor integrity and promotion, or inhibition of salvage pathways managing the stress response, may induce tumor cell death pathways. Whereas non-tumor cells compensate for therapeutically induced perturbations, as indicated by the modest toxicity profile of editing trials (Table 1) [132,133]. Tumor tissue editing techniques finally inhibit the relief of stress in tumor tissue, which cancer cells are relying on for survival. This way, tumor tissue editing approaches seem to resolve or attenuate M-CRAC.

Only combined transcriptional regulation with nuclear receptor agonists or cytokines (interferon-α) led either to decisive inflammation control and cCR, e.g., in r/r renal clear cell carcinoma, r/r LCH and r/r HL, or to differentiation induction in AML, or to the establishment of non-oncogene addiction in r/rHL [12,17,25,45]. 

The clinical results demonstrate that combined transcriptional modulation on-topic unlocks the phenotypic plasticity of tumor tissues for M-CRAC control (Figure 1 and Figure 2). Unlocking is profoundly associated with the reprogramming of cancer hallmarks in edited tumor tissues, on the background of the reprogrammed tumor system’s stress response, which facilitates a tighter phenotype at the phenotypically and genetically heterogeneous tumor sites. The tight phenotype might be an explanation for why tumor tissue editing may overcome genetic heterogeneity at different metastatic tumor sites and associated resistance. 

The combination of both treatment components, metronomic low-dose chemotherapy plus transcriptional modulation, is profoundly associated with the reprogramming of cancer hallmarks to biologic hallmarks controlling M-CRAC in respective edited tumor tissues (Figure 2). 

The prerequisite of exceptionally low doses of metronomic chemotherapy for clinically efficacious transcriptional modulation reveals a novel aspect of the mechanisms of action of metronomic chemotherapy. Both treatment components, low-dose metronomic chemotherapy and transcriptional modulation, provide poor or no monoactivity in r/r tumors [40,55]. 

Concerted reprogramming of cancer hallmarks with metronomic (very) low-dose chemotherapy and dual/triple transcriptional modulation induces a series of clinically important phenomena beyond M-CRAC control. These are tumor cell death, edition of non-oncogene addiction, maintenance of disease control beyond discontinuation of editing therapy, or re-establishment of hormone sensitivity in CRPC [14,19,37].

## 7. Repurposing Chemotherapy: Metronomic Low-Dose Chemotherapy

Metronomic chemotherapy is clinically efficacious despite frequently applied scheduled dose reductions (Table 1). Clinical efficacy of ‘very’ low-doses within tissue editing protocols demonstrates the meaning of ‘low-dose’ metronomic chemotherapy. Thus, the clinical biomodulatory effect of metronomic low-dose chemotherapy is not necessarily based on the cumulative achievement of maximum tolerable doses with equally split daily doses applied during a 3 to 4-week treatment cycle. 

A main result of the presented series of tissue editing trials is that metronomic ‘very’ low-dose chemotherapy is sufficient and essential for reprogramming of the oncogenic stress response and for transcriptional tumor editing. Metronomic low-dose chemotherapy may concretely contribute to stimulating the immune response, particularly in concert with additional immunomodulators (immune checkpoint inhibitors, pioglitazone). Metronomic low-dose chemotherapy may inhibit tumor-associated inflammation, e.g., in combination with pioglitazone, interferon-α, and angiogenesis, and/or may promote tumor differentiation, e.g., in combination with all-trans retinoic acid. Finally, metronomic low-dose chemotherapy facilitates edited non-oncogene addiction, namely the successful use of repurposed targeted therapies, e.g., mTOR inhibitors, associated with resolution of M-CRAC in r/r Hodgkin’s lymphoma and consecutive cCR [13,17,45,57]. 

Low-dose metronomic chemotherapy, as used in tissue editing approaches, represents a completely repurposed way of applying chemotherapy. Alkylating agents or methotrexate have been frequently applied with respect to their immunomodulatory activity profile (Table 1) [57]. 

Reprogramming the tumor-associated stress response, the novel activity profile of metronomic chemotherapy, determines the therapeutic difference in comparison to the well-known mechanisms of action observed following pulsed chemotherapy. Prolonged therapy-free intervals for recovering toxicity, the risk of infections, and M-CRAC development are well-known problems associated with conventional chemotherapy [134]. Focusing on tumor cell death with maximum tolerable doses always bears the risk of post-therapy M-CRAC induction due to evolving molecular-genetic and genetic tumor heterogeneity and drug resistance.

Tissue editing approaches integrating low-dose metronomic chemotherapy induce, as shown, not only replication arrest or dormancy, associated with stable disease or stable objective response, even beyond discontinuation of tissue editing, but also CR and cCR in r/r neoplasias (Table 1) [19,25]. 

Thus, the overall impact of low-dose metronomic chemotherapy is not simply to save treatment costs in countries that cannot afford expensive targeted therapies or to integrate a mostly non-approved therapy into ‘Common Sense Oncology’ [135,136]. Rather, metronomic chemotherapy may address the weaknesses of pulsed chemotherapies, targeted therapies, oncogene-directed, or non-oncogene ‘addicted’ therapies by targeting the tumor’s plasticity via phenotypic editing of undruggable genetic defects in tumor cells, for example, in AML patients with complex chromosomal aberrations, including defects in gatekeeper genes such as TP53 mutations and PTEN downregulation, which may be counteracted with pioglitazone [13,45,100]. Gatekeeper defective genes, e.g., aberrant expression of PTEN could probably be normalized, as clinically suggested in r/r Hodgkin’s lymphoma, by edited non-oncogene addiction to mTOR [13,137]. 

## 8. Examples of M-CRAC Control and Tissue Editing in the Clinical Setting

In a large series of studies, tissue editing approaches were applied for the treatment of relapsed or refractory tumor diseases of quite different histologic origins, e.g., carcinoma, sarcoma, and hematologic neoplasias (Table 1) [12,13,14,15,16,17,18,19,20,21,22,23,24,25,26,27,28]. A recent randomized trial compared in r/r metastatic non-small cell lung cancer (NSCLC) (second-to-fifth-line treatments) such as two biomodulatory therapy approaches, nivolumab at the approved dose level versus metronomic low-dose chemotherapy, pioglitazone, COX-2 inhibitors, and clarithromycin. The experimental arm failed to show superiority in progression-free survival (PFS). Nevertheless, the fact that overall survival (OS) was similar between the treatment arms gives rise to the hypothesis that the well-tolerable biomodulatory therapy may edit and prime tumor tissues for efficacious, consecutive immune checkpoint inhibitor (ICPi) therapy in the experimental arm [23,138,139].

In r/r Hodgkin’s lymphoma, two cohorts of patients are retrospectively summarized (Figure 3) [13,27]. Patients had either all received, at that time, approved systemic treatments or were ineligible for standard treatment, including immune checkpoint inhibitors. One patient received a previous allogenic hematopoietic stem-cell transplant (alloHSCT). The rescue therapy consisted of an all-oral tissue editing approach including metronomic, daily, low-dose treosulfan, pioglitazone, etoricoxib, and dexamethasone, plus, in repurposed edited tissue, everolimus 15 mg p.o. daily to achieve serum trough levels of 15 ng/mL (MEPED schedule for r/r HL). Everolimus has some but poor monoactivity in r/r Hodgkin’s disease [140]. All patients achieved continuous, complete remission with MEPED. After entering CR, three transplant-eligible patients underwent consolidating alloHSCT and achieved consecutive cCR. 

As shown, in Hodgkin’s lymphoma, both lymphoma cell death and M-CRAC control may be achieved with an on-topic guided therapeutic procedure using inflammation control, enhancement of immunosurveillance, and metabolic reprogramming [55,141]. An earlier small series of patients showed that response to treatment positively correlated with systemic inflammation control [27].

The addition of everolimus to pulsed standard chemotherapy in r/r HL failed to show any beneficial clinical effect [142]. Therefore, the clinical data reveal that tissue editing effects in r/rHL are associated with an edited non-oncogene addiction to mTOR [142] (Figure 1).

Multisystem Langerhans cell histiocytosis (mLCH) is an inflammation-driven malignant hematologic disease with strong PPARγ expression on malignant histiocytes [27,143]. A concerted anti-inflammatory, immunomodulatory, and metabolic reprogramming approach in r/r multi-system LCH with low-dose trofosfamide, etoricoxib, pioglitazone, and low-dose dexamethasone led to long-term control; cCR was observed in cases with involvement of the pituitary gland or meningeal involvement [16,27].

A tissue editing approach tailored to the treatment of AML comprising low-dose azacitidine (AZA), pioglitazone, and all-trans retinoic acid (ATRA) could induce hematologic or molecular complete remission in non-promyelocytic leukemia (non-PML) AML by inducing differentiation of blasts to granulocytes, which has also been corroborated in vitro (Figure 3) [12,44,50]. The differentiated leukemic blasts may gain functional importance, as suggested by the resolution of fungal pneumonia during therapy. Differentiation induction in AML blasts might provide a broad pattern of cell death pathways, like in neutrophils [144]. As in r/r Hodgkin’s lymphoma, relapse following alloHSCT may be rescued in AML by differentiation induction [45,47]. 

Response duration at different organ sites (skin versus bone marrow) was organ-dependent in AML [46]. Recent negative results of ATRA in addition to low-dose chemotherapy underline the important therapeutic impact of pioglitazone and azacitidine in addition to ATRA for M-CRAC control [145]. 

Differential tumor tissue editing approaches for overcoming M-CRAC: The four chosen examples of r/r tumor neoplasias treated via tumor tissue editing techniques demonstrate the successful use of quite different editing approaches. All approaches facilitate M-CRAC control in r/r tumor disease via concerted inflammation control in r/r LCH and r/r HL, differentiation induction in r/r AML with adverse risk features according to European LeukemiaNet (ELN), and the efficacious targeting of edited non-oncogene addiction in r/r HL and melanoma with mTOR inhibitors [12,13,16,23,41,45] (Figure 3). In r/r NSCLC, an immunomodulatory effect of the editing schedule may be suggested due to the observed efficacy of rescue therapy with immune checkpoint inhibitors [23].

Thus, from a therapeutic point of view, the resolution or attenuation of the clinically defined M-CRAC phenomenon is highly differentially accessible despite a uniquely arising therapeutic problem, M-CRAC. 

Clinical data on successful tumor tissue editing in r/r neoplasias descriptively separate M-CRAC and, therefore, provide an example for knowledge-generating patient care. Experimental studies on tumor tissue editing must still describe in more detail the concrete mechanisms of action responsible for M-CRAC control (Figure 2 and Figure 4). However, it is clinically obvious that tumor tissue editing techniques may overcome M-CRAC. 

## 9. Addressing M-CRAC Control with a Novel Therapy Model for r/r Neoplasias 

Tumor tissue editing, as clinically demonstrated, promotes therapeutic access to system-relevant normative structures and functions (Figure 4) and reverses tumor-promoting normative functions, such as hallmarks of cancer, into biologic hallmarks controlling tumor growth or inducing cCR in r/r tumor disease. The clinically accessible surrogates, such as inflammation control, differentiation induction, establishment of non-oncogene addiction, enhancement of immunosurveillance, or the reprogramming of tumor metabolism (pioglitazone) reveal the possibility of selecting differential editing approaches even within a histologic tumor type. The clinically monitored surrogates that indicate the reprogramming of cancer hallmarks do not sufficiently cover the suggested activity profiles of the used transcriptional modulators [55]. For example, pioglitazone participates in modulating metabolic processes and immunosurveillance. Alternative apoptosis pathways, like ferroptosis, may be initiated with pioglitazone [146].

The activity profiles of the chosen biomodulatory drugs in differential tissue editing protocols may be context-dependently specified, as impressively shown for pioglitazone [55]. Pioglitazone and metronomic low-dose chemotherapy are the backbone in all histologically quite different r/r neoplasias (Table 1). Therefore, tumors share intra- and extracellular communication protocols. Tissue editing protocols provide unique accessibility in r/r hematologic and oncologic neoplasias and are the basis of context-dependently targeting M-CRAC (Table 1). 

The novel tumor model is currently primarily based on patient-reported outcomes and clinically evaluable outcome parameters, such as the correlation of inflammation and M-CRAC control, differentiation induction, establishment of non-oncogene addiction, or enhancement of immune surveillance [23]. Alternative ways of tumor cell death induction may be suggested from the known mechanisms of action of single drugs used in editing schedules [55]. 

In the future, the huge repertoire of biomodulatory active drugs could expand, dependent on novel editing-relevant surrogates, the reprogramming of cancer hallmarks and ways of oncogenic stress response. Accordingly, novel biomodulatory combinations may be established and drug repurposing exploited, particularly with respect to edited non-oncogene addiction (Table 1, Figure 1) [37,147]. Molecular imaging, imaging-based 3D invasion culture, imaging mass cytometry and ‘omic’ technologies could be pivotal techniques for studying mechanisms involved in providing therapeutically relevant tumor plasticity and for guiding the appropriate selection of biomodulatory active drugs addressing the M-CRAC problem (Figure 4) [148,149,150,151,152].

The new therapy model represents a substantial advancement as complete remission or continuous complete remission may be achieved in histologically quite different r/r neoplsias, i.e., r/r non-PML AML, RCCC, angiosarcoma, cholangiocellular carcinoma, HL, and mLCH [13,16,24,25,26,45]. If long-term disease control is the best response to tissue editing protocols, competitive classic maintenance or rescue therapies are available, particularly in tumors with druggable driver mutations or high tumor mutational burden [23,153,154,155]. However, a general advantage of tissue editing approaches is modest toxicity, as exemplified in a randomized trial for r/r hepatocellular carcinoma in comparison to nivolumab [23]. Classic maintenance therapies may also show interesting biologic effects, such as an impact on consecutive progression-free survival (PFS2), e.g., in multiple myeloma, or an improvement in overall survival, as exemplarily shown for non-small cell lung cancer [153,154].

## 10. Patient Selection Criteria for Treatment with Tumor Tissue Editing Approaches

Regulatory authorities cannot approve tumor tissue editing approaches due to missing phase III trials. The first randomized phase III trial by Patil et al. might lead to the approval of metronomic chemotherapy plus a reduced dose of nivolumab combined with erlotinib in relapsed head and neck cancer. The presented trials are phase I and II trials for relapsed and refractory neoplasias. Thus, only individual treatment concepts can be an indication for tumor tissue editing approaches. The formal establishment of differential tumor tissue editing techniques urgently requires larger phase II or III trials. 

## 11. Conclusions and Future Directions

The oncogenic potential of systemic tumor therapies, as reflected by the promotion of metastases, tumor repopulation, acquired cancer cell resistance, and genetic heterogeneity (M-CRAC), poses a crucial barrier to long-term survival in metastatic cancer. Successful patient care in refractory or relapsed neoplasias with a unique therapeutic principle, tumor tissue editing, facilitated clinically separating the oncogenic potential of M-CRAC as a phenomenon that arises as a post-therapeutic disease. Tumor tissue editing successfully resolves M-CRAC or bypasses M-CRAC with differentiation induction (Table 1). Moreover, clinical data on M-CRAC control additionally reveal that heterogeneous types of macro-metastasis and organ parenchymal interfaces promoting tumor invasion may be controlled or resolved by tissue editing techniques [156].

Tumor tissue editing may achieve both tumor cell death and M-CRAC control by using on-topic guided editing techniques. These techniques encompass differentiation induction, inflammation control, enhancement of immunosurveillance, and metabolic reprogramming with pioglitazone [55,141,157]. Resistance mechanisms, developing in response to qualitatively quite different preceding systemic or local radiotherapies, may be controlled by editing techniques, as exemplified in r/r AML and r/r Hodgkin’s lymphoma [12,13,44,45,46,47].

Weiss et al. identified shared nodules promoting metastasis and therapy resistance in tumors and suggested new opportunities to improve tumor therapy with novel therapeutic strategies [11]. Clinical trials on tissue editing in relapsed or refractory tumor diseases with different histologies now indicate that the novel biomodulatory strategies may overcome significant obstacles to long-term disease control induced by pulsed standard therapies. 

Successfully introduced tissue editing schedules reveal that editing must be specifically adapted according to tumor histology. Nevertheless, histologically different neoplasias share common patterns of targets for the unique therapeutic reprogramming of cancer hallmarks with respective differential recombination of transcriptional modulators (Table 1). Stress response to low-dose metronomic chemotherapy is suggested to be a prerequisite for the differential combined activity of transcriptional modulators or for the concertedly edited stress response, referred to as non-oncogene addiction [133,158]. 

Tissue editing identifies M-CRAC as a unique disease trait by reversing tumor promoting hallmarks of cancer into biologic hallmarks, controlling, resolving, or bypassing M-CRAC. The clinical results give hints on mechanisms of action as indicated by clinical surrogates, inflammation control, differentiation induction and establishment of non-oncogene addiction. The shared ‘tissue nodes’ promoting M-CRAC are objects for further research. 

The novel therapy model addressing M-CRAC control may initialize multileveled, clinically linked studies on tumor tissue editing for deciphering communicative networks that provide tumor-inherent therapeutic plasticity as a prerequisite for successful editing (Figure 4). Biomodulatory drug combinations may therapeutically disturb the tumor’s novel, evolved homeostatic mechanisms for managing the stress response [132]. Targeting tumor cell survival with tumor tissue editing techniques shows promise for principally controlling M-CRAC and for inducing tumor cell death with biomodulatory therapeutic techniques.

Designing tissue editing therapies is just the beginning, and novel concepts may include quite different drugs, including immunotherapeutics. Immunotherapy could be applied in novel settings to promote synergistic potency and efficacy with novel biomodulatory tumor tissue reprogramming, i.e., ‘pro-anakoinotic’ therapy schedules [40,57]. 

In addition, the control of the M-CRAC disease traits following classic induction therapies warrants the evaluation of differential tissue editing approaches. The introduction of M-CRAC control in future therapeutic considerations may help overcome the multifold translational challenges of precision medicine by repurposing targeted therapies. Such therapy approaches could be beneficial for the very large group of r/r tumor diseases without driver mutations [159,160,161,162].

## Figures and Tables

**Figure 1 cancers-16-00180-f001:**
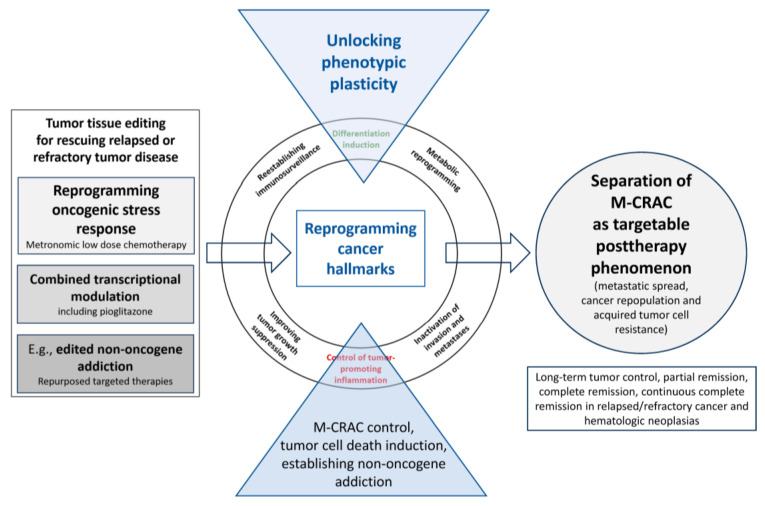
Tumor tissue editing for rescuing relapsed or refractory tumor disease separates M-CRAC as targetable phenomenon: suggested mechanisms of action.

**Figure 2 cancers-16-00180-f002:**
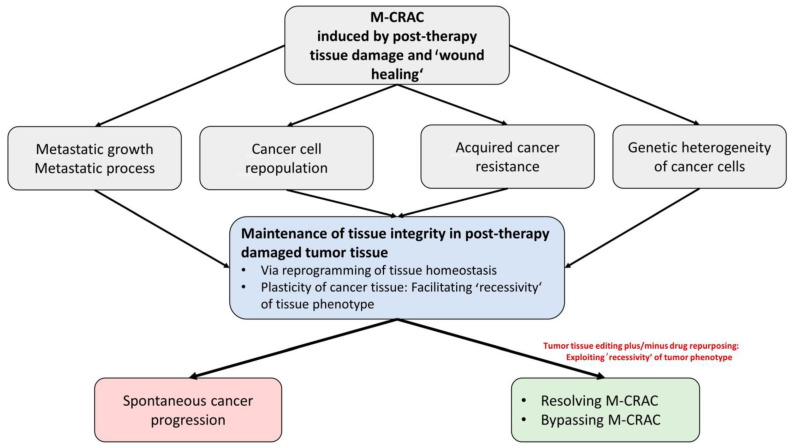
Disease traits of M-CRAC trigger spontaneous cancer progression. The therapeutic tumor tissue editing tool may control, resolve, or bypass M-CRAC via access to shared triggers for tumor promotion present in histologically different tumor types.

**Figure 3 cancers-16-00180-f003:**
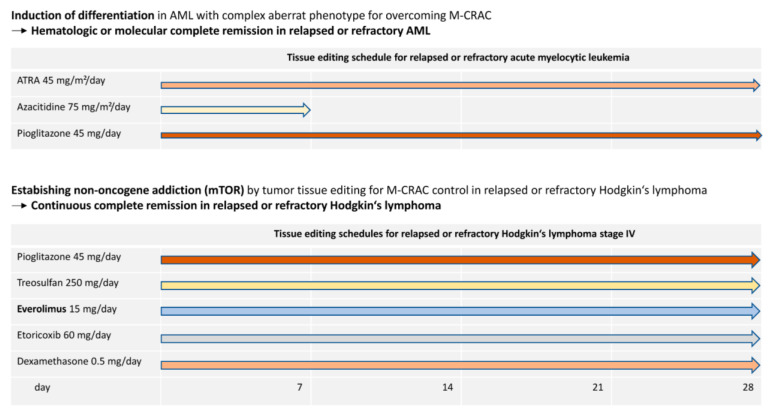
Differential tumor tissue editing approaches for overcoming M-CRAC.

**Figure 4 cancers-16-00180-f004:**
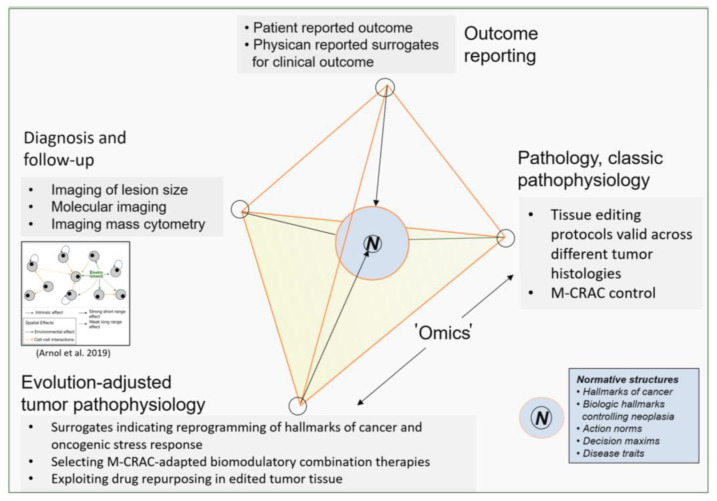
A novel therapy model: tumor tissue editing for M-CRAC control derived from knowledge-generating patient care with tumor tissue editing techniques. Arnol et al. drafted a communication-derived tumor model [40,48].

**Table 1 cancers-16-00180-t001:** Tumor tissue editing protocols for relapsed or refractory tumors: transcriptional modulation plus low-dose metronomic chemotherapy (MCT) for M-CRAC control and resolution (CR: complete remission, cCR: continuous complete remission). PR = partial remission. SD= stable disease. ®: randomization. n.d = not done.

Transcriptional Regulation	MCT, Targeted Therapy	r/r Neoplasia [12,13,14,15,16,17,18,19,20,21,22,23,24,25,26,27,28,29,30]	Reprogramming Cancer Hallmarks	Best Response	Reference Citation
Pioglitazone, rofecoxib	MCT	Hepatocellular carcinoma	Inflammation control	PR	[20]
Pioglitazone, rofecoxib	MCT	Cholangiocellular carcinoma	n.d.	cCR	[26]
Pioglitazone, rofecoxib	MCT	High-grade gliomas	n.d.	SD	[31]
Pioglitazone, rofecoxib	MCT	Angiosarcoma	n.d.	cCR	[24]
Rofecoxib plus/minus pioglitazone	MCT	Metastatic gastric cancer	® n.d.	PR	[21]
Rofecoxib plus/minus pioglitazone Pioglitazone, etoricoxib	MCT MCT + temsirolimus	r/r metastatic melanoma uveal melanoma	® Inflammation control	PR SD	[22,41]
Pioglitazone, etoricoxib	MCT, clarithromycin vs. nivolumab	r/r Non-small cell lung cancer	® Enhancing immune surveillance	PR	[23]
Pioglitazone, dexamethasone, etoricoxib	MCT MCT + imatinib	Castration-refractory prostate cancer	n.d.	PR	[18,30,42]
Pioglitazone, dexamethasone, etoricoxib	MCT, everolimus	r/r Hodgkin’s lymphoma	Inflammation control	cCR	[13,27]
Pioglitazone, dexamethasone, etoricoxib	MCT	r/r Multisystem Langerhans cell histiocytosis	Inflammation control	cCR	[16,28]
Pioglitazone, rofecoxib Pioglitazone, rofecoxib, + interferon-α	MCT MCT	r/r Renal clear cell carcinoma r/r Renal clear cell carcinoma	No sufficient inflammation control Inflammation control	SD cCR	[17,25,43]
Pioglitazone, dexamethasone	MCT, lenalidomide	r/r Multiple myeloma	n.d.	PR	[14]
Pioglitazone, all-trans retinoic acid	Azacytidine	r/r Non-promyelocytic acute myelocytic leukemia	Differentiation induction	CR	[44,45,46,47]

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
