# Peer review of "Addressing Genetic Tumor Heterogeneity, Post-Therapy Metastatic Spread, Cancer Repopulation, and Development of Acquired Tumor Cell Resistance"

_cancers, 2023, doi:10.3390/cancers16010180_

Round 1

Reviewer 1 Report

Comments and Suggestions for Authors

By describing the concept of M-CRAC as a major contributor to resistance to conventional cancer therapies and discussing tissue editing as a potential therapeutic approach, this Research Topic details new avenues for addressing therapeutic issues arising from refractory or relapsed tumors. . Although the review provides an in-depth summary and analysis of relevant content, certain issues remain to be addressed.

1. Please briefly describe the emergence of the term “tumor tissue editing” in the literature and its adoption as a new therapeutic paradigm.

2. The authors should conduct a comparative analysis of traditional chemotherapy regimens and metronomic low-dose chemotherapy, emphasizing the benefits of the latter.

3. Please describe in detail how tumor tissue editing methods redirect cancer-related markers into biomarkers.

4. I would like to know how tissue editing approaches impact the genomic evolution of tumors, particularly in terms of acquired mutations and clonal selection, and discuss the implications for long-term disease control.

5. Clarify whether tissue editing approaches impact genomic evolution of tumors, particularly with respect to acquired mutations and clonal selection, and discuss potential implications for long-term disease control.

6. Are there standardized patient selection criteria for treatment using tumor tissue editing?

Comments on the Quality of English Language

There are many long sentences in the article, which are relatively obscure and difficult to understand. Writers should consider breaking complex sentences into smaller, more digestible units. This will enhance the readability and make it easier for readers to follow the argument

Author Response

Reviewer 1:

First of all, we would like to thank the reviewer for taking the time to review our manuscript. Tracked changes mode was used to visualize changes.

  1. Please briefly describe the emergence of the term “tumor tissue editing” in the literature and its adoption as a new therapeutic paradigm.

We added a corresponding paragraph ll 89-92.

  1. The authors should conduct a comparative analysis of traditional chemotherapy regimens and metronomic low-dose chemotherapy, emphasizing the benefits of the latter.

We included a paragraph showing the benefits of metronomic low-dose chemotherapy in comparison to conventional chemotherapy ll. 405-438

  1. Please describe in detail how tumor tissue editing methods redirect cancer-related markers into biomarkers.

We added a detailed paragraph on how tumor tissue editing methods redirect cancer-related markers into biomarkers ll 150-180

  1. I would like to know how tissue editing approaches impact the genomic evolution of tumors, particularly in terms of acquired mutations and clonal selection, and discuss the implications for long-term disease control.

AND

  1. Clarify whether tissue editing approaches impact genomic evolution of tumors, particularly with respect to acquired mutations and clonal selection, and discuss potential implications for long-term disease control.

We added a detailed paragraph on how tumor tissue editing methods redirect cancer-related markers into biomarkers and we discussed potential implications for long-term disease control ll 181-205

  1. Are there standardized patient selection criteria for treatment using tumor tissue editing?

A discussion of patient selection criteria for treatment using tumor tissue editing was added ll 559-565

There are many long sentences in the article, which are relatively obscure and difficult to understand. Writers should consider breaking complex sentences into smaller, more digestible units. This will enhance the readability and make it easier for readers to follow the argument.

Several long sentences were shortened to enhance readability.

Reviewer 2 Report

Comments and Suggestions for Authors

In this review, Harrer DC and collaborators analysed M-CRAC in cancer progression. As a phenomenon arising as posttherapy disease, M-CRAC is involved in promotion of metastases, tumor repopulation, acquired cancer cell resistance, and genetic heterogeneity. Tissue editing approaches, such as metronomic low-dose chemotherapy or the use of transcriptional modulators plus/minus targeted therapies, could achieve tangible clinical benefit. Moreover, tissue editing approaches orchestrates a multi-pronged approach simultaneously engaging tumor cell differentiation, immunomodulation, and inflammation control, which are important drivers of M-CRAC in a multitude of different tumor.

Comments/questions:

Overall, this manuscript is interesting and well organized.

1)     As a crucial component of tumor microenvironment, which is the role of myeloid-derived suppressive cells, TAMs, CAFs and tumor stroma-components on M-CRAC. The role of immunomodulation on M-CRAC should be expanded.

2)     English editing or style should be revised. For example, lines 244-246 should be checked.

Comments on the Quality of English Language

 Moderate editing of English language is required

Author Response

Reviewer 2:

First of all, we would like to thank the reviewer for taking the time to review our manuscript. Tracked changes mode was used to visualize changes.

1)     As a crucial component of tumor microenvironment, which is the role of myeloid-derived suppressive cells, TAMs, CAFs and tumor stroma-components on M-CRAC. The role of immunomodulation on M-CRAC should be expanded.

A paragraph on tumor microenvironment (e.g., myeloid-derived suppressive cells, TAMs) was added ll 341-352

2)     English editing or style should be revised. For example, lines 244-246 should be checked.

Extensive corrections were made to enhance the readability of this manuscript.

Round 2

Reviewer 1 Report

Comments and Suggestions for Authors

The author has answered my questions very well and I have no more questions.